# Elastic anisotropy differentiation of thin shale beds and fractures using a novel hybrid rock physics model

Haoyuan Li*[1], Xuri Huang[1], Lei Li[2], Fang Li[2] and Tiansheng Chen[3]

[1]School of Geosciences and Technology, Southwest Petroleum University, Chengdu, 610500, China
[2]Research Institute of Hainan Branch Company, China National Offshore Oil Corporation,570300, China
[3]Sinopec Petroleum Exploration and Production Research Institute, Beijing, 100083, China

*Correspondence to*: Haoyuan Li (cqjtu.geo.lee@foxmail.com)

**Abstract.** Elastic anisotropy is frequently used to characterize fracture distribution. However, sets of parallel horizontal fractures and thin shale beds in tight sand both can cause vertical transverse isotropy. Here, we are not referring to shale layers on the logging scale but rather to very thin shale beds, a few centimeters thick, within tight sand. To accurately differentiate the anisotropy caused by horizontal fractures or thin shale beds, we propose a hybrid rock physics model. This new model combines the Hudson model and the shale compacting Orientation Distribution Function (ODF) model, based on the anisotropic Self-Consistent Approximation and Differential Effective Medium (SCA&DEM) theory. The new model's reliability is demonstrated by comparing to the well logs. The proposed model can characterize the elastic properties of both thin shale beds and horizontal fractures. Based on this model, the rock physical analysis reveals that thin shale beds and horizontal fractures exhibit distinct elastic anisotropy characteristics. Furthermore, we analyse the seismic response differences between horizontal fractures and thin shale beds using the anisotropic Ruger's approximation formula. The analysis indicates that the seismic response of tight sand containing thin shale beds interferes with the fracture's identification. On the other hand, there are identifiable differences between the fractured tight sand and the tight sand containing thin shale beds. Based on this difference, we develop a new seismic attribute to characterize the fracture distribution. These difference-based attributes can effectively eliminate the interference from thin shale beds, making the distribution of horizontal fractures more apparent.

## 1 Introduction

With the continuous growth of global energy demand and the gradual depletion of conventional oil and gas resources, the development of unconventional oil and gas resources has become increasingly important (Gharavi et al., 2023). Among these unconventional resources, fracture reservoirs have become a research focus due to their excellent storage and flow capacity (Zhang et al., 2022). The presence and development of fractures affect the permeability and porosity of reservoirs and directly related to the occurrence and flow of oil and gas (Liu et al., 2019; Jiang et al., 2023). Therefore, accurately predicting fracture distribution is crucial for oil and gas exploration and development. In recent years, fracture prediction techniques based on seismic data have made significant progress, becoming an important tool for fracture containing gas and oil research (Liu et al., 2018; Li et al., 2020). For example, the application of seismic bright spot technology has proven effective in practice for fracture containing gas prediction (Fawad et al., 2020). The main idea of this technology is that fractures exhibit a significant low impedance contrast with the adjacent rock formations. However, thin shale beds present in tight sand may also exhibit the same low impedance properties. These two situations can cause similar seismic responses, which can mislead the

characterization of fracture distribution. Therefore, it is crucial to differentiate the elastic properties caused by the fractures or thin shale beds (Lin et al., 2022). It is worth noting that the thin shale beds we studied are very thin, with thicknesses of a few centimetres, which are far below the resolution of well logging. Therefore, it is difficult to describe these thin shale beds within tight sand using seismic and logging techniques. Consequently, we need to rely on rock physics elastic models to equivalently represent their microstructure and convert it into macroscopic responses at seismic and logging scales.

The rock physics elastic modelling process is categorized into three primary components: matrix, skeleton, and fluid Mavko (2020). The matrix model represents the amalgamation of the diverse minerals found in a rock based on their composition. For a homogeneous mineral matrix, various averaging methods can be used to synthesize an isotropic rock matrix. Voigt (1890) proposed the equivalent strain-averaging model. Reuss (1929) proposed the equivalent stress-averaging model. These models give the theoretical elastic parameters range of the rock. Hill (1952) gave the elastic parameters by averaging the upper and lower bounds. Wyllie and Gregory (1953) proposed a linear formula so that, when the rock has uniformly distributed intergranular pores, there is a linear relationship between porosity and acoustic transit time. Hashin and Shtrikman (1963) gave the lower bound of elastic combination parameters for the softest rock and the upper bound for the hardest rock of the mineral composition. By using the above methods, the mineral component equivalent medium is treated as the rock matrix (Alabbad et al., 2023).

The skeleton models are rock structure models with inhomogeneous phases inserted into the matrix background phase (Ma et al., 2024). Kuster and Toksöz (1974) gave a skeleton-equivalent model for different pore shapes in carbonate. Hudson (Hudson, 1980, 1981) proposed a flat coin-shaped crack equivalent model. Schoenberg and Sayers (1995) gave a linear-slip model. Xu and White (1995, 1996) combined the self-consistent model to optimize the Kuster and Toksöz model. Xu and Payne (2009) further gave an equivalent pore model for carbonate. Chapman et al. (2010) gave a multi-scale fractures equivalent model. Lian et al. (2024) combined experimental measurement results to obtain the compacted optimized skeleton model using compaction coefficients.

The fluid models describe the elastic and anisotropic characteristics of various fluids in relation to the skeleton. For simple isotropic models, Gassmann (1951) gave the fluid replacement formula at low frequencies. Brown and Korringa (1975) proposed an anisotropic Gassmann formula (B&K model) for anisotropic skeleton (Thomsen, 2023). Using a fluid substitution model, Guo et al. (Guo et al., 2023) investigated the relationship between acoustic velocity and fluid saturation. However, the microstructure of thin shale beds is complicated (Zhou et al., 2023). Hence, it is necessary to give greater attention to the properties of elasticity and anisotropy. Therefore, thin shale beds should not be regarded as a component of a matrix model or as a straightforward skeletal inclusion phase. This research aims to integrate the anisotropic model of thin shale beds with a specific component of the rock skeleton. For the orientation of the thin shale beds, Roe (1965) defined the direction functions in the three-dimensional space of rock. These direction functions can be represented by the Legendre coefficients corresponding to a sequence of functions. For the thin shale beds by compaction, their normal is parallel to the third axis, so it can be equivalent to the VTI (Vertical Transverse Isotropic) medium. For VTI media, Sayers (1995) simplified the Legendre

coefficients into two, which can express the elastic stiffness matrix of the thin shale beds. On the other hand, Johansen et al. (2004) gave the SCA and DEM models to calculate stiffness matrix of the single thin shale bed.

In this paper, we first propose a model to combine the thin shale beds and horizontal fracture skeleton. This model is verified by field data in a Sichuan Basin gas reservoir. The tight sand containing fractures and the thin shale beds are funded in this area (Ding et al., 2021; Yurikov et al., 2021). Based on the fracture orientations and dip angles within the area, we assume the fracture anisotropy to be VTI anisotropy. Then we analyze the seismic response differences between fracture and thin shale beds using the anisotropic Ruger's approximation formula. Finally, based on the seismic response, we develop a new seismic attribute to explore the potential position of the horizontal fractures.

## 2 Method

### 2.1 Rock physics modelling process

To study the VTI anisotropy of shale and horizontal fractures, we proposed the modelling workflow shown in Figure 1. (Vertical fractures, due to their orientation, exhibit HTI anisotropy, which can be directly distinguished from shale without modelling.) The modelling process is mainly divided into two parts: the fractures skeleton and the thin shale beds rock physics modelling. Specifically, we calculated the stiffness matrix of the sand skeleton containing fractures and fluids in the sand model. On the other hand, we computed the stiffness matrix of the beds structure in the thin shale beds model. Finally, we combined the two stiffness matrices using the SCA and DEM model to obtain the rock stiffness matrix that includes horizontal fractures and thin shale beds. The method of obtaining the parameters required for the whole modelling process and their meanings are shown in Table 1.

Table 1 The meaning and method of the rock model parameters. The table outlines the methods for obtaining the parameters required in the technical process depicted in Figure 1, along with their corresponding definitions. These fundamental parameters facilitate the utilization of rock physics elastic models, enabling the conversion between elastic properties and the other physical parameters.

| Name | Method and Way | Meaning |
| --- | --- | --- |
| Mineral contents | Logging, SEM and XRD | The composition of the matrix |
| Minerals modulus | Sonic Measurement | The elastic properties of minerals |
| Fluid contents | Logging | The composition of the fluid |
| Fluid modulus | Sonic Measurement | The elastic properties of fluid |
| Aspect ratio of pore | Thin section analysis, SEM and XRD | The skeletal structure |
| Critical porosity | Logging and Sonic Measurement | The rock structure |

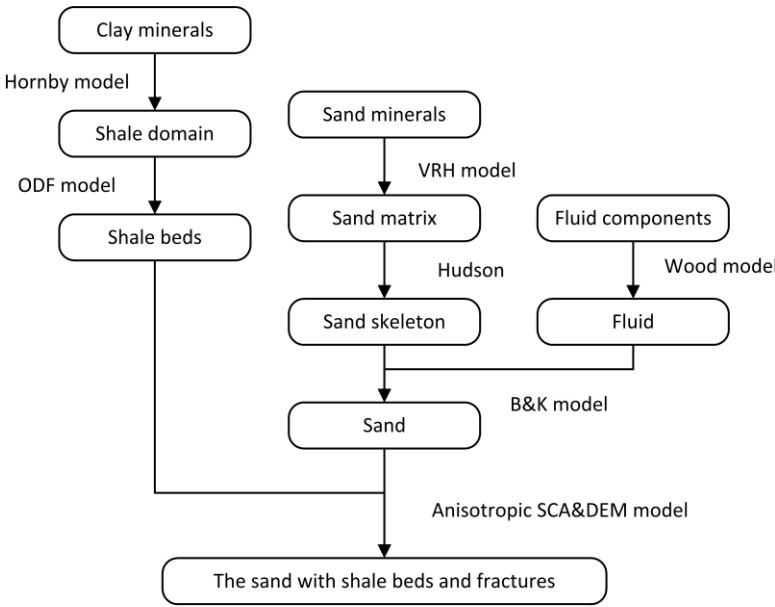

**Figure 1 The rock modelling flowchart. The modelling consists of three branches. The left branch simulates the elastic properties of shale using the Orientation Distribution Function (ODF) and Hornby models. The middle branch focuses on the elastic properties of the fracture with the Hudson approach. right branch simulates the elastic properties of fluids based on the Wood model.**

### 2.1.1 the fracture skeleton rock physics modelling process

For the fracture skeleton, we used the VRH model (Hill, 1952) to build the sand matrix (*Msand*):

$$Msand = \left( \sum_{i=1}^{N} f_i M_i + \frac{1}{\sum_{i=1}^{N} \frac{f_i}{M_i}} \right) / 2,$$  ( 1 )

where $f_i$ is the content of the i-th mineral in the matrix, $M_i$ is the isotropic modulus of i-th mineral in the matrix (bulk modulus $Ki$ or shear modulus $\mu i$). $Msand$ is background isotropic modulus (bulk modulus $Km$ or shear modulus $\mu m$). The isotropic modulus can be converted into the sand isotropic stiffness of the matrix using the following equation:

$$\boldsymbol{C}_{sandm} = \begin{bmatrix} Km + \frac{4}{3}\mu m & Km - \frac{2}{3}\mu m & Km - \frac{2}{3}\mu m & 0 & 0 & 0 \\ Km - \frac{2}{3}\mu m & Km + \frac{4}{3}\mu m & Km - \frac{2}{3}\mu m & 0 & 0 & 0 \\ Km - \frac{2}{3}\mu m & Km - \frac{2}{3}\mu m & Km + \frac{4}{3}\mu m & 0 & 0 & 0 \\ 0 & 0 & 0 & \mu m & 0 & 0 \\ 0 & 0 & 0 & 0 & \mu m & 0 \\ 0 & 0 & 0 & 0 & 0 & \mu m \end{bmatrix},$$  ( 2 )

Hudson model (Hudson, 1980) is based on a scattering-theory analysis of the mean wave field in an elastic solid with thin, penny-shaped ellipsoidal cracks or inclusions. He proposed a Taylor expansion approximation to calculate the stiffness matrix for fracture-porosity composite system (Hudson et al., 1996):

$$C_{dsand} = C_{sandm} + C^1 + C^2 ,$$ (3)

where $C^1$ is the first-order correction term for the anisotropy caused by the fracture, and $C^2$ is the second-order correction term of the anisotropy caused by the mutual coupling between the directional fractures. Both $C^1$ and $C^2$ are calculated from the pore aspect ratio a, the rock matrix porosity $\phi m$ and fracture porosity $\phi f$ ($\phi = \phi m + \phi f$). We assume that the percentage of fracture porosity to total porosity $\phi$ in the reservoir is a constant (3.2%). These parameters of skeleton were obtained from log curves and thin sections.

We introduced fluids into the sand skeleton using the BK model (Brown and Korringa, 1975) to obtain the saturated sand rock stiffness matrix:

$$s_{ijkl}^{sand} = s_{ijkl}^{dsand} - \frac{(s_{ijaa}^{sand} - s_{ijaa}^{sandm})(s_{bbkl}^{sand} - s_{bbkl}^{sandm})}{(s_{ccdd}^{sand} - s_{ccdd}^{sandm}) - \phi\left(\frac{1}{K_{fl}} - \frac{1}{K_{sandm}}\right)},$$ (4)

where the parameters $s_{ijkl}^{dsand}$ and $s_{ijkl}^{sandm}$ represent the flexibility of dry rock skeleton and rock matrix minerals respectively. The stiffness matrix can be inverted from flexibility matrix following $C_{ijkl} S_{ijkl} = I$ and vice versa. $K_{fl}$ can be obtained by the Wood (1956) formula.

**2.1.2 the thin shale beds rock physics modelling process**

The thin shale beds in the tight sand can be considered as composed of clay domains (Bandyopadhyay, 2008). These clay domains exhibit laminar structures and strong anisotropy. To construct a single clay domain, we referenced Hornby's procedure (Hornby et al., 1995). Hornby's method effectively describes the complex structure of a single clay domain. After we get the anisotropy of a single clay domain, the equivalent elastic stiffness of shale beds' orientation is obtained by taking the Voigt average (Sayers, 1995):

$$C_{shale\_11} = L + 2M + \frac{4\sqrt{2}}{105}\pi^2\left[2\sqrt{5}a_3 W_{200} + 3a_1 W_{400}\right],$$ (5)

$$C_{shale\_33} = L + 2M \frac{16\sqrt{2}}{105}\pi^2\left[\sqrt{5}a_3 W_{200} - 2a_1 W_{400}\right],$$ (6)

$$C_{shale_{12}} = L - \frac{4\sqrt{2}}{315}\pi^2\left[2\sqrt{5}(7a_2 - a_3)W_{200} - 3a_1 W_{400}\right],$$ (7)

$$C_{shale_{13}} = L + \frac{4\sqrt{2}}{315}\pi^2\left[\sqrt{5}(7a_2 - a_3)W_{200} - 12a_1 W_{400}\right],$$ (8)

$$C_{shale_{44}} = M - \frac{2\sqrt{2}}{315}\pi^2\left[\sqrt{5}(7a_2 + a_3)W_{200} + 24a_1 W_{400}\right],$$ (9)

$$C_{shale\_66} = \frac{C_{shale\_11} - C_{shale\_12}}{2}.$$ (10)

Where, $a_1, a_2, a_3, L$ are the elastic parameters of a single clay domain ($a_1 = C_{claym_{11}} + C_{claym_{33}} - 2C_{claym_{13}} - 4C_{claym_{44}}$, $a_2 = C_{claym_{11}} - 3C_{claym_{12}} + 2C_{claym_{13}} - 2C_{claym_{44}}$ , $a_3 = 4C_{claym_{11}} - 3C_{claym_{33}} - C_{claym_{13}} - 2C_{claym_{44}}$ , $L = \frac{1}{15}\left(C_{claym_{11}} + C_{claym_{33}} + 5C_{claym_{12}} + 8C_{claym_{13}} - 4C_{claym_{44}}\right)$ and $c_{claym}$ is the elastic matrix of a single clay domain. The

coefficients W200 and W400 are the Legendre coefficients of Orientation Distribution Function (ODF) and can be obtained through X-ray diffraction (XRD) experiments on core samples. However, for shale, core measurement is challenging and expensive. Thus, we used the compaction distribution function $W(\xi)$ derived by Johansen (2004) to calculate these two parameters as following:

$$W_{200} = \sqrt{\frac{5}{2}} \int_{-1}^{1} W(\xi) P_2(\xi) d\xi, \tag{11}$$

$$W_{400} = \sqrt{\frac{9}{2}} \int_{-1}^{1} W(\xi) P_4(\xi) d\xi, \tag{12}$$

where

$$P_2(\xi) = \frac{1}{2}(3\xi^2 - 1),$$

$$P_4(\xi) = \frac{1}{8}(35\xi^4 - 30\xi^2 + 3),$$

$$\xi = \cos(\theta),$$

$$W(\xi) = \frac{1}{4\pi} \frac{A^2}{(\xi^2 + A^2(1-\xi^2))^{\frac{3}{2}}}.$$

The only parameter that needs to be input is A (compaction factor), which represents the ratio of shale thicknesses before and after compaction. It is worth noting that obtaining A is difficult. For more convenient applications, we will use the critical porosity-based compacted ODF to characterize the beds orientation. Specifically, the relationship between A and the critical porosity is given by (Bachrach, 2011):

$$A = \left(\frac{\phi}{\phi_0}\right)^k. \tag{13}$$

The model parameter k controls the speed at which the pore space deforms (Bachrach, 2011). Thus, the value of k is calibrated based on the shale compaction curve, and for normal compaction, k equals 1. In the entire process of shale modelling, the sole free variable, A, can be converted into the relationship between porosity and critical porosity. Critical porosity is defined as the point at which clay particles in suspension make contact, leading to a phase transition that results in a finite shear modulus.

### 2.1.3 Thin shale beds and fracture skeleton hybrid rock physics model

The stiffness matrices of thin shale beds and fracture skeleton were calculated using the previous two sections, and the thin shale beds were inserted into the fracture skeleton model using the SCA and DEM model. To avoid having sand and shale as two isolated entities, the model constructs the structure in two steps. First, establish a mixed medium with half sand and half shale, then randomly insert parts with actual sand or shale content exceeding half. The specific steps are as follows:

First, the anisotropic SCA model (Hornby et al., 1995) is used to construct a shale and sand bi-connected equivalent structure. The bi-connected equivalent structure is:

$$C_{shale-sand}^{bio} = \sum_{n=1}^{2} 0.5 C_n \left(I + \hat{G}(C_n - C^{bio})\right)^{-1} \left\{\sum_{p=1}^{2} 0.5 \left(I + \hat{G}_{ijkl}(C_n - C^{SCA})\right)^{-1}\right\}^{-1}. \tag{14}$$

Here, when n and p are equal to 1, it refers to the shale stiffness matrix, and when they are equal to 2, it refers to the sand stiffness matrix. And $C_{shale-sand}^{bio}$ is the stiffness matrix of the shale and sand bi-connected equivalent structure. The anisotropic DEM model is then used to complete the remaining structure. $\hat{G}_{ijkl}$ represents the geometric parameters of the inclusions, and its calculation process is provided by Mura (2013). The insufficient components are added gradually in equal amounts to the shale and sand bi-connected equivalent structure until the actual shale content of the rock is reached. Thus, the calculation of the DEM model is an iterative process. Specifically, it involves subdividing the inclusions into $n$ parts. With each addition of a part, the stiffness matrix of the background phase is updated. The result of the $i$-th iteration is as follows:

$$\frac{d\left(C^{DEM}(v_i)\right)}{dv_i} = \frac{1}{(1-v_i)}\left(C^{inclusion}(i) - C^{DEM}(v_i)\right)\left[\underline{I} + \hat{G}_{ijkl}\left(C^{inclusion} - C^{DEM}(v_i)\right)\right]^{-1}, \tag{15}$$

$$C^{DEM}(v_1) = C_{shale-sand}^{bio}, \tag{16}$$

$$C^{DEM}(v_{i+1}) = C^{DEM}(v_i) + d\left(C^{DEM}(v_i)\right), \tag{17}$$

$$C^{DEM}(v_n) = C^{rock}. \tag{18}$$

Where $v_i$ and $C^{DEM}(v_i)$ are respectively the corresponding inclusion volume content and elastic matrix of the inclusion component before the i-th insertion. $C^{inclusion}$ is the stiffness matrix of the inclusion component, and $C^{rock}$ is the stiffness matrix of the result.

The final rock stiffness matrix can be used to obtain the rock acoustic and anisotropic parameters through the velocity-elasticity relationship and the anisotropy model (Thomsen, 1986):

$$V_p = \sqrt{\frac{c_{33}^{rock}}{\rho}}, \tag{19}$$

$$V_s = \sqrt{\frac{c_{44}^{rock}}{\rho}}, \tag{20}$$

$$\varepsilon = \frac{c_{11}^{rock} - c_{33}^{rock}}{2c_{33}^{sat}}, \tag{21}$$

$$\gamma = \frac{c_{66}^{rcok} - c_{44}^{rock}}{2c_{44}^{rock}}, \tag{22}$$

$$\delta = \frac{\left(c_{13}^{rock} + c_{44}^{rock}\right)^2 - \left(c_{13}^{rock} - c_{44}^{rock}\right)^2}{2c_{33}^{rock}\left(c_{33}^{rock} - c_{44}^{rock}\right)}, \tag{23}$$

where $\rho$ , $V_p$, $V_s$ are the equivalent density, velocities of the longitudinal wave and the shear wave for the rocks respectively. $\varepsilon$, $\gamma$, $\delta$ are the Thomsen anisotropy parameters. And the equivalent density can easy calculate by Voight's averaging.

# 3 Background and Model calibration

## 3.1 Geological background

The research focuses on the tight sand gas in the Xujiahe Formation in the Sichuan Basin. The region has dense lithology, with fractures serving as the primary migration pathways (Huang et al., 2022). Horizontal fractures can help identify tight gas reservoirs in this area (Yue et al., 2018; Zhang, 2021). The developed horizontal fractures can also effectively assist in the water injection development of tight gas (Zhao et al., 2021). Previous Chinese scholars have analysed horizontal fractures, which exhibit preferential distribution and structural features of VTI anisotropy (Su, 2011). The core samples and imaging logging showed horizontal fractures in this study region (Figure 2). We also statistically analyzed the fracture distribution primarily based on the imaging logging and core samples.

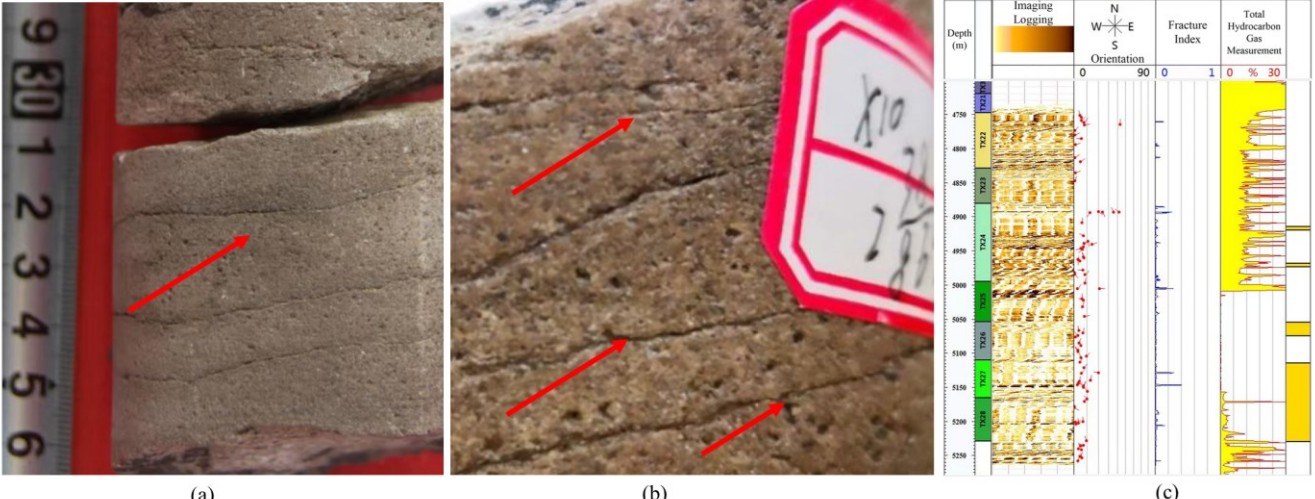

**Figure 2 The evidence for the horizontal fractures in the area. Figure 2a and 2b are cores with developed horizontal fractures (The red arrows point to the horizontal fractures.). Figure 2c is the imaging logging of the horizontal fracture formation.**

Through the core samples, we have analyzed the width and length of these horizontal fractures (Figure 3a, 3b). Most fractures have a width of less than 1 mm, with lengths primarily ranging from 100 to 300 mm. Therefore, we set the fracture aspect ratio a in the Hudson model to 0.01. We also calculated the orientation of these horizontal fractures though imaging logging (Figure 3c). Most fractures were concentrated around the 60° direction. It revealed that these horizontal fractures exhibited VTI anisotropy. Therefore, we used the VTI equivalent model to study the anisotropy of fractures in the area. Han et al.(2022) analysed the fracture VTI anisotropy in the area, but did not consider the VTI anisotropy caused by thin shale beds.

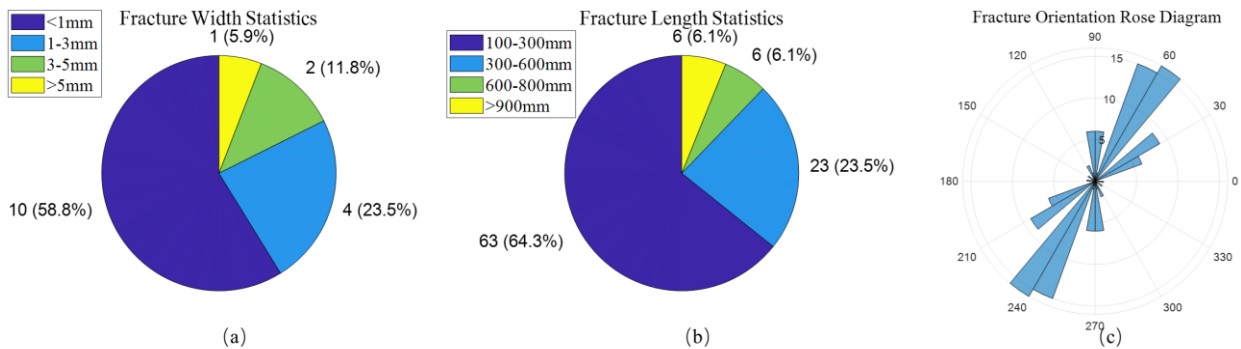

**Figure 3 Fracture parament statistics. The 3a and 3b show the statistics of fracture length and width, respectively. Most fractures have a width of less than 1 mm, with lengths primarily ranging from 100 to 300 mm. The figure 3c is a rose diagram of fracture orientations, showing that most fractures are concentrated around the 60° direction.**

We conducted statistical analysis on the average fracture porosity and total porosity interpretation results from 10 wells in the study region. The results are shown in the figure 4. The fracture porosity in the region ranges from 0.04% to 0.39%, accounting for 3.2% of the total porosity.

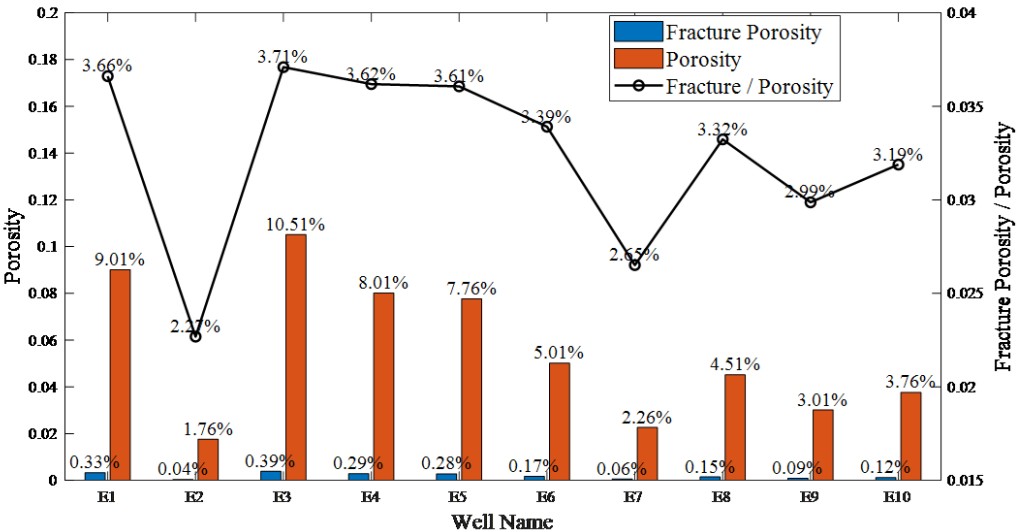

**Figure 4 Fracture porosity and matrix porosity analysis. The bar chart represents the average values of porosity and fracture porosity within the target interval from logs in the area. The line chart indicates the percentage of fracture porosity relative to total porosity.**

The compliance effectively characterizes a material's ability to deform under stress, making it particularly suitable for representing fractures. The Linear Slip Deformation (LSD) model proposed by Schoenberg and Sayers(1995) assumes a linear relationship between the displacement discontinuity across a fracture surface and the applied stress, enabling the compliance contributions from multiple fractures to be directly summed. In contrast, Hudson's model treats fractures as

small perturbations to the stiffness tensor, based on the assumption that fractures introduce only minor modifications to the medium. Specifically, it uses a first-order Taylor expansion to approximate the effects of fractures on the stiffness. This approach makes stiffness perturbation a more suitable and computationally efficient framework for modeling low fracture

densities. Furthermore, stiffness-based models offer a more direct approach for analyzing seismic wave propagation, particularly when fracture porosity is sufficiently low. When fracture porosity exceeds 0.45% (fracture density = 0.1), the Hudson model becomes less effective in describing the elastic properties of fractures (Figure 5). In this study, the fractures exhibit a maximum porosity of 0.39%, which is well within the descriptive capabilities of the Hudson model. Therefore, we adopted the Hudson model for our analysis.

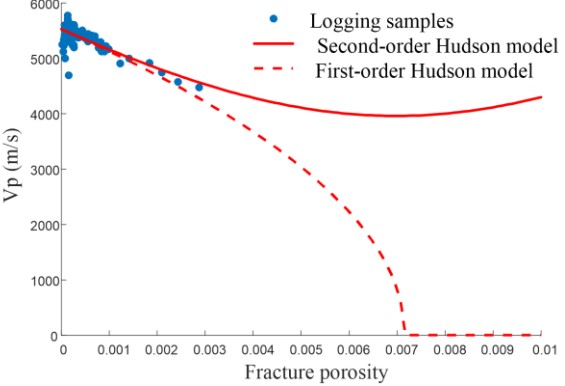

**Figure 5 The solid line represents the first-order Hudson formula, the dashed line represents the second-order Hudson formula, and the points indicate well log samples. The first-order results of the Hudson model consider the effects of fractures, while the second-order results account for both the effects of fractures and their interactions. In this study, we used the second-order results of the Hudson model.**

The area is characterized by the widespread development of tight sand containing thin shale beds (Figure 6), which interferes with our prediction of fracture zones (Wu et al., 2022). These compacted shales exhibit a microstructure with a preferential orientation in the plane (Bandyopadhyay, 2008), which can be confused with fracture-induced anisotropy.

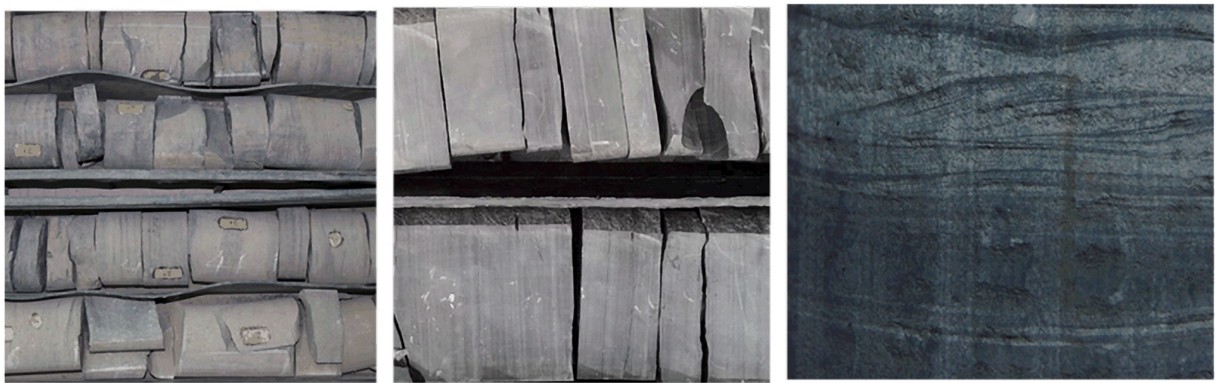

**Figure 6 The tight sand containing thin shale beds. The light-colored sections of the core is tight sand, while the dark-striped sections**
**are thin shale beds.**

## 3.2 Model calibration

To verify and calibrate the background parameters of the model, we analyze the sample points within the well. We extracted P-wave and S-wave velocity under three different control variable environments, as shown in Figure 7. Figures 7a, 7b show the effect of shale content on the velocity of rock for saturation between 0.6 and 0.8 and porosity between 0 and 0.02, respectively. Figures 7c, 7d show the effect of saturation on the velocity of the rock when the shale content is between 0.09 and 0.11 and the porosity is between 0 and 0.02, respectively. Figures 7e 7f show the influence of porosity on the velocity of rock when the saturation is between 0.6 and 0.8 and the shale content is between 0.09 and 0.11, respectively.

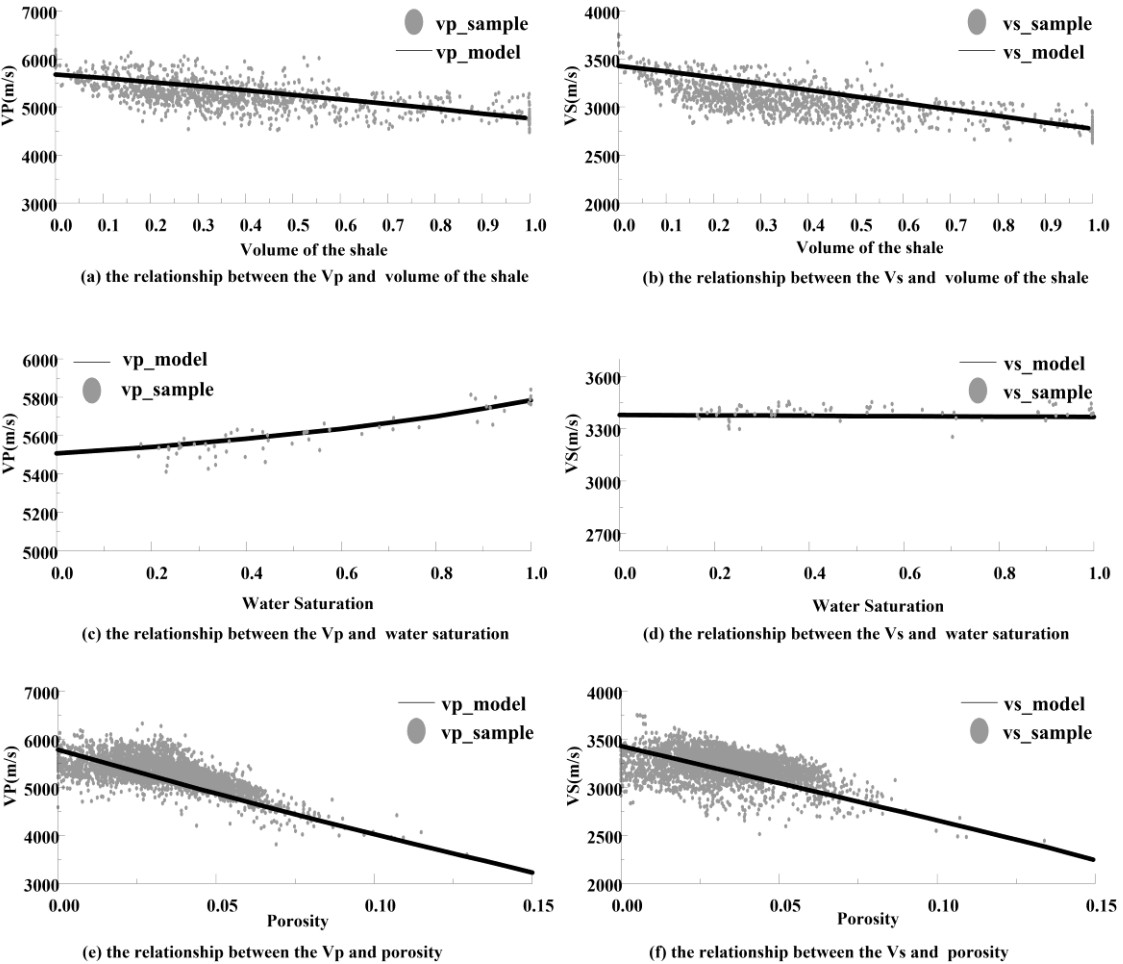

**Figure 7 The model results compared with actual well logging measurements. 7a and 7b are the effect of shale content on velocity. 7c and d are the effect of saturation on velocity. 7e and 7f are the influence of porosity on velocity.**

As previously mentioned, the physical properties in this area are complex, so sample points exhibit some divergence with the hybrid model. However, the variation trends in physical property are consistently with the model under different conditions. We can utilized these trends to calibrate the model's background parameters (Table 2).

**Table 2 Table of Background Parameters for Rock Physics Modelling. These data primarily originate from the appendix of "The Rock Physics Handbook"(Mavko et al., 2020) and represent the most commonly used fundamental rock parameters in the field.**

| | Bulk modulus (GPa) | Shear modulus (GPa) | Density (g/cm$^3$) | Vp (km/s) | Vs (km/s) |
|---|---|---|---|---|---|
| Clay | 25 | 9 | 2.55 | 3.81 | 1.88 |
| Quartz | 36.6 | 45 | 2.65 | 6.05 | 4.15 |
| Feldspar | 37.5 | 15 | 2.62 | 4.68 | 2.39 |
| Water | 2.56 | 0 | 1.05 | 1.5 | 0 |
| Gas | 0.038 | 0 | 0.23 | 0.34 | 0 |

## 4 Analysis and Application

In this chapter, we focused on the impact of the thin shale beds and fractures on the elastic anisotropy. Based on the petrophysical characteristics of tight sand in the area, three theoretical models were established (Table 3). The fracture parameters were set using the parameters defined in Section 3.1, while the background parameters were derived from Table 2. These models were used to verify the elastic anisotropy of different types of tight sand. The model 1 represents the tight sand containing fractures. The model 2 means tight sand containing fractures and thin shale beds. The model 3 refers to the tight sand containing thin shale beds.

The main physical property of a rock is porosity, which indicates whether the rock has enough space to collect and migrate fluids. Therefore, we analyzed the effect of porosity on the acoustic velocity and Thomsen parameters of the three models, as shown in Figure 8. Figures 8a, 8b, and 8c depict the effect of porosity on rock acoustic velocity. The elastic parameters of fractures and thin shale beds decrease with increasing porosity, with Model 1 (fractures) being more sensitive to velocity changes than the other models. On the other hand, Figures 8d, 8e, and 8f depict the anisotropic characteristics of the rock. The anisotropy parameter increases with increasing porosity, especially in Model 1. From the analysis of porosity, we can see that thin shale beds and fractures have a similar trend in their influence on acoustic velocity and anisotropy, but the sensitivity of these elastic characteristics is greater in fractures than in thin shale layers.

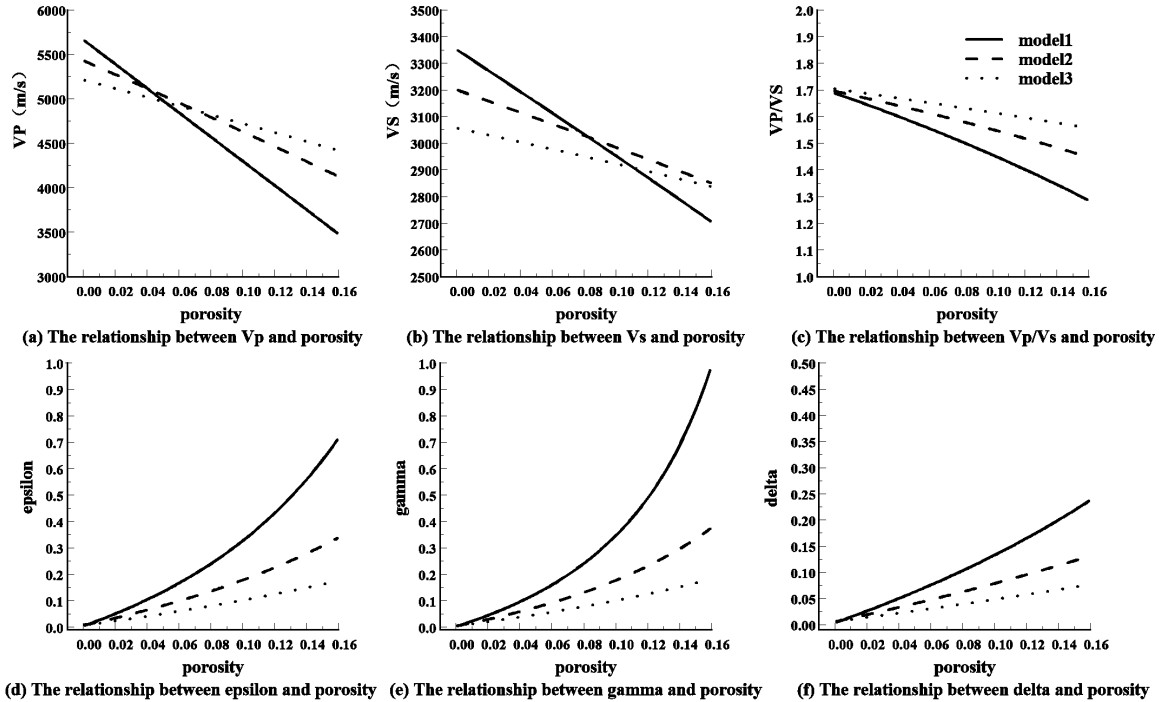

**Figure 8 The impact of porosity on the three models. Figure 8a to 8c are the impact of porosity on rock acoustic velocity. Figure 8d to f are the anisotropic characteristics of the rock influenced by porosity. In the plots, the black solid line represents Model 1 (fractures model), the dashed line represents Model 2 (hybrid model), and the dotted line represents Model 3 (shale model).**

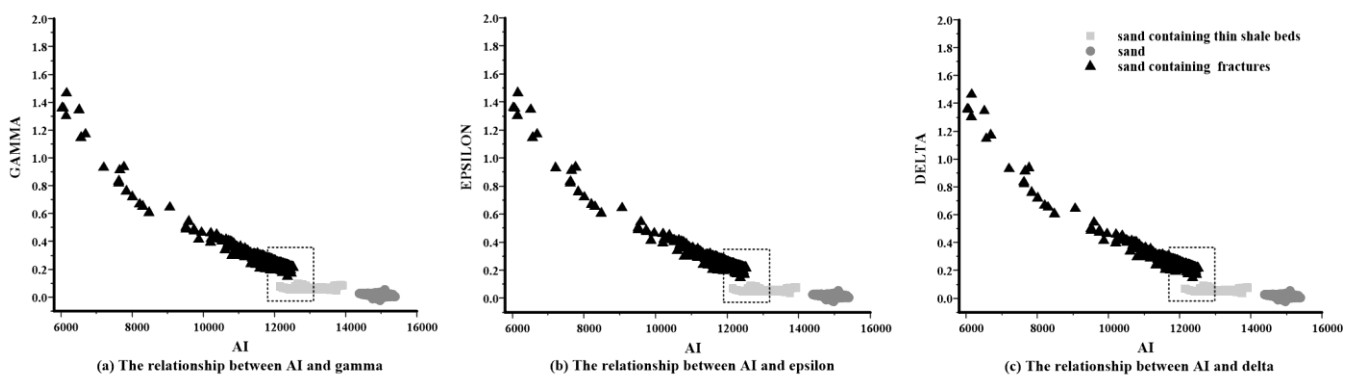

**Figure 9 The anisotropic results of sample points in the working area. All logging samples were collected from the target and adjacent sections, with classifications based on logging data. In the figure, the black triangles represent tight sand containing fractured, the dark gray circles represent tight sand without anything, and the light gray circles represent tight sand containing thin shale beds. The dashed box highlights sample where the two types of tight sand have similar acoustic impedance.**

Therefore, we further analyze the Thomsen anisotropy parameters variation with acoustic impedance of the three corresponding theoretical models existed in the well logs. Tight sand containing fractures have different Thomsen anisotropy

from tight sand without fractures. On the other hand, tight sand containing thin shale beds and fractures have the same acoustic impedance within the dashed box but vary differently in anisotropy parameters (Figure 9). This means that the best way to distinguish them is pre-stack inversion. Applying most methods based on post-stack seismic data remains challenging. To facilitate the subsequent description, we define the portion of tight sand containing fractures that have the same response as tight sand containing thin shale beds as "complex sand."

**Table 3 The parameters of the three models. The model parameters listed in the table are established based on statistical analysis of well log and geological data. Model 1 corresponds to tight sand containing fracture, Model 2 represents hybrid sand, and Model 3 pertains to sand containing thin shale beds.**

| Name | | Model1 | Model2 | Model3 |
|---|---|---|---|---|
| Mineral Composition | VSH (%) | 20 | 40 | 60 |
| | VFS (%) | 40 | 30 | 20 |
| | VQU (%) | 40 | 30 | 20 |
| | PHI (%) | 15 | 10 | 5 |
| Pore | AR | 0.01 | 0.05 | 0.1 |
| | Critical Phi (%) | 40 | 40 | 40 |
| Fluid | SW (%) | 40 | 50 | 60 |

To further investigate the pre-stack seismic response characteristics of the sand containing them, we utilize the anisotropic Ruger approximation formula proposed by Wang (2024) to analyze the amplitude variation with the incident angle for both. We apply a three-layer model with different elastic and anisotropy parameters listed in table 4.

**Table 4 The parameters in the table are derived from calibrated rock physics models. The first two rows represent the parameters for a background layer, while the remaining rows represent the calculated elastic parameters under various degrees of fracture development in the target layer.**

| Layer (Thickness 50 m) | Vp(m/s) | Vs (m/s) | Density(g/cm3) | Delta | Gamma | Epsilon |
|---|---|---|---|---|---|---|
| Upper layer(Background) | 5700 | 3485 | 2.61 | 0 | 0 | 0 |
| Bottom layer(Background) | 5700 | 3485 | 2.61 | 0 | 0 | 0 |
| Containing thin shale beds | 4754 | 2763 | 2.55 | 0.019 | 0.02 | 0.009 |
| Containing fractures (pore 0.02) | 5514 | 3622 | 2.59 | 0.0717 | 0.0372 | 0.0654 |
| Containing fractures (pore 0.06) | 4691 | 3313 | 2.53 | 0.263 | 0.1235 | 0.2657 |
| Containing fractures (pore 0.1) | 3912 | 3011 | 2.46 | 0.553 | 0.2298 | 0.6593 |
| Containing fractures (pore 0.14) | 3205 | 2712 | 2.40 | 1.01 | 0.36 | 1.676 |

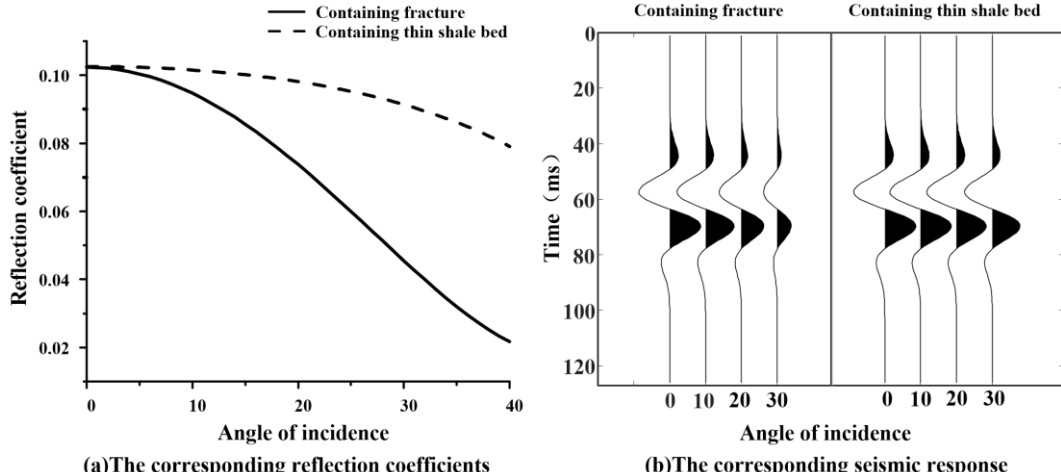

(a)The corresponding reflection coefficients  (b)The corresponding seismic response

**Figure 10 The pre-stack seismic angle response characteristics in the dashed box. Figure 10a is the relationship between reflection coefficient and incidence angle. The solid line represents the results for tight sand containing fractures (from the dashed box in Figure 9), and the dashed line represents the results for tight sand containing thin shale layers. At smaller incidence angles, the reflection coefficients of both tight sands are similar. Figure 10b is the synthetic seismic records obtained from the convolution of the reflection coefficients.**

The pre-stack seismic angle gather and corresponding reflectivity coefficient are shown in Figure 10. The forward analysis shows that when the incident angles are small, the reflection coefficients are close, the waveforms are similar. However, as the incident angle increases, the complex sand reflectivity decays faster, and the waveform is weakened more significantly compared with tight sand containing thin shale beds.

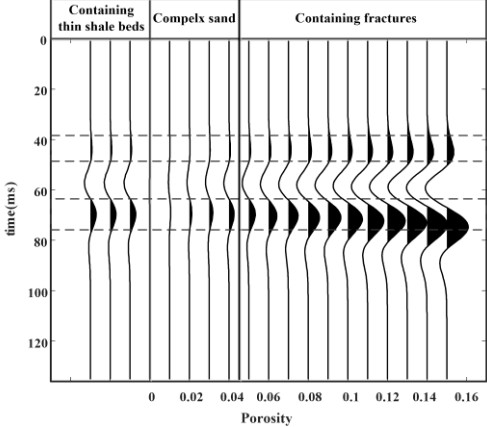

**Figure 11 The response corresponding to different porosities in tight sand (right) compared with the response of tight sand with shale beds (left).**

In this paper, considering the post seismic data are easily obtained and with small data size. We propose the fractures and thin shale beds distinguish method based on the post seismic data and previous hybrid rock physics model. we analyze the post-stack seismic response of tight sand based on their physical properties. Excitingly, complex sand corresponds to the low-

porosity portion of tight sand containing fractures. This means that tight sand with high porosity is not considered complex sand and significantly differs from both complex sand and tight sand containing thin shale beds (Figure 11).

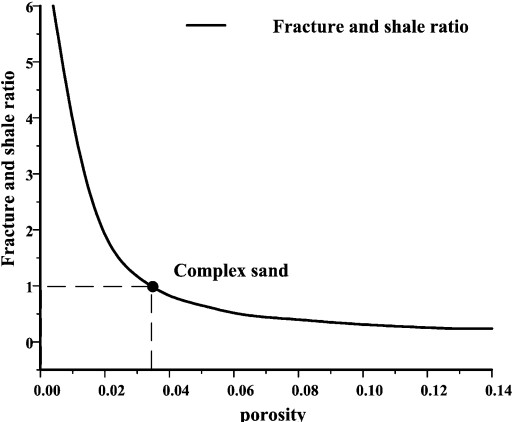

**Figure 12 the Fracture and shale ratio of waveforms in tight sand varying with porosities. The attribute value of 1 corresponds to the complex sand within the dashed box in Figure 8 with lower attribute values indicating higher porosity (more developed fractures).**

To further quantify the waveform differences, we extracted the ratio $r_1$ of the maximum peak amplitude and peak travel time between 60 ms and 90 ms as a waveform shape attributes of the tight sand containing fractures. Similarly, we extracted the
315 ratio $r_2$ of the maximum peak amplitude and peak travel time between 35 ms and 50 ms as a waveform shape attributes of the tight sand containing thin shale beds. We define the ratio parameter $\frac{r_1}{r_2}$ (namely fracture and shale ratio) to describe the similarity between the tight sand containing thin shale beds and containing fractures. As shown in Figure 12, when the porosity is between 0.025 and 0.03, the ratio value approach 1, which means that the response of the tight sand containing thin shale beds and containing fractures is similar, indicating complex sand. As mentioned earlier, this difference-based new attribute
(ratio parameter) can effectively identify the tight sand containing fractures. The quantified results show that ratio values are less than 0.8. Thus, the differences in waveforms between the two tight sands can be used to describe the potential range of the tight sand containing fractures. We calculate the attribute of anisotropic aspect ratio of the target layer. We first select a tight sand containing shale beds layer as the reference layer to facilitate the evaluation of differences between the target layer and the tight sand containing shale beds layer. The dashed line in figure 13 represents the tight sand containing shale beds
layer (reference layer), the solid black line represents the target layer. For analysis, we extracted anisotropic aspect ratio attributes from the two layers.

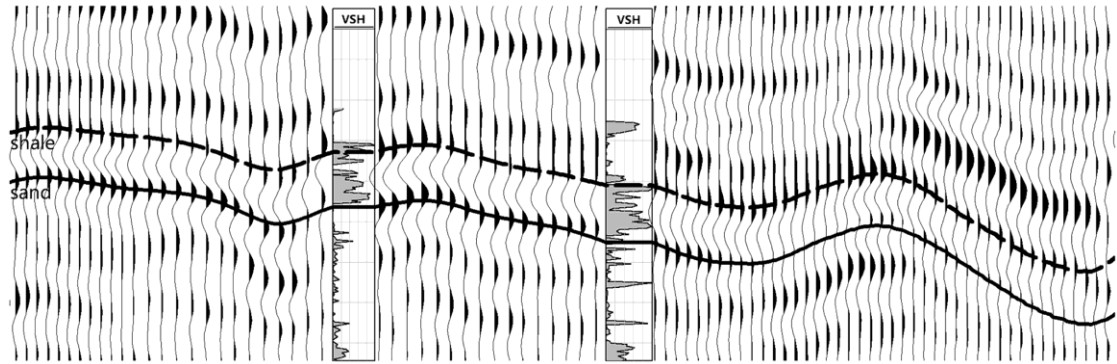

**Figure 13 Profile extracted from two tight sand layers. The solid line represents the target layer where fracture evaluation is required, and the dashed line represents the reference layer which containing thin shale beds, identified through logging and geological analysis.**

The results along the target layer are shown in Figure 13. As previously discussed, the larger the difference, the smaller the ratio, and the more developed the fractures are. Conversely, ratios close to 1 or greater than 1 indicate dense rocks or shale-containing dense sandstones. Therefore, in the figure 14, black areas represent well-developed fractured dense sandstones, while white areas represent tight sand containing low porosity or thin shale beds. The new seismic attribute shows a clearer correlation with fault distribution, which are depicted with red lines in figure 13.

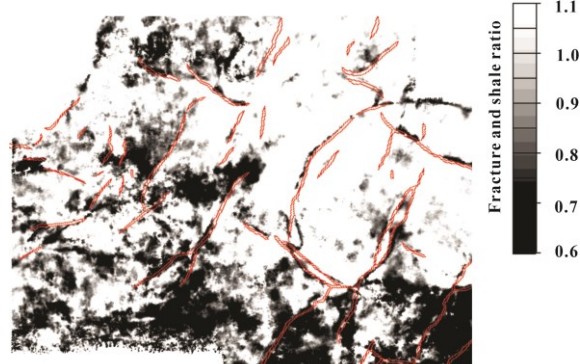

**Figure14 Fracture and shale ratio map. In the figure, dark areas represent regions with high fracture content, while light and gray areas indicate tight sand or tight sand containing thin shale beds. Red lines mark fault lines.**

## 5 Discussion

Simultaneously studying fractures and thin shale beds in tight sand presents significant challenges. Previous research primarily focused on either fractures or shale, but not both together. As discussed in Chapter 3, current techniques for identifying fractures in tight sand using seismic data are limited, particularly in formations with thin shale beds. In Chapter 2, we emphasized methods that couple the elastic characteristics of thin shale beds with tight sand skeletons. Complex models, which include various experimental and derived approaches, often require more computational resources and input parameters than

simpler methods like the Gassmann model, especially in processes such as forward and inverse modelling based on rock physics. Despite these complexities, complex models offer advantages in capturing more nuanced aspects of rock behaviour and properties. To address these computational demands, we have proposed a novel approach. During our theoretical model analysis, we summarized a seismic attribute based on the differences between fractures and shale beds. In our subsequent discussions, we will further evaluate both the reliability and the limitations of our proposed methodology compared to existing

technologies.

## 5.1 Comparison with other shale models

Current shale models utilized in logging and seismic analysis predominantly rely on SCA and DEM models, yet these fail to consider the preferred orientation of shale plates. This limitation arises due to the inherent complexity in mathematically expressing shale plate orientation and the formidable challenge of measuring the corresponding model parameters. In tight

sand containing thin shale beds, the orientation of shale plates significantly influences the elastic properties of the formation.

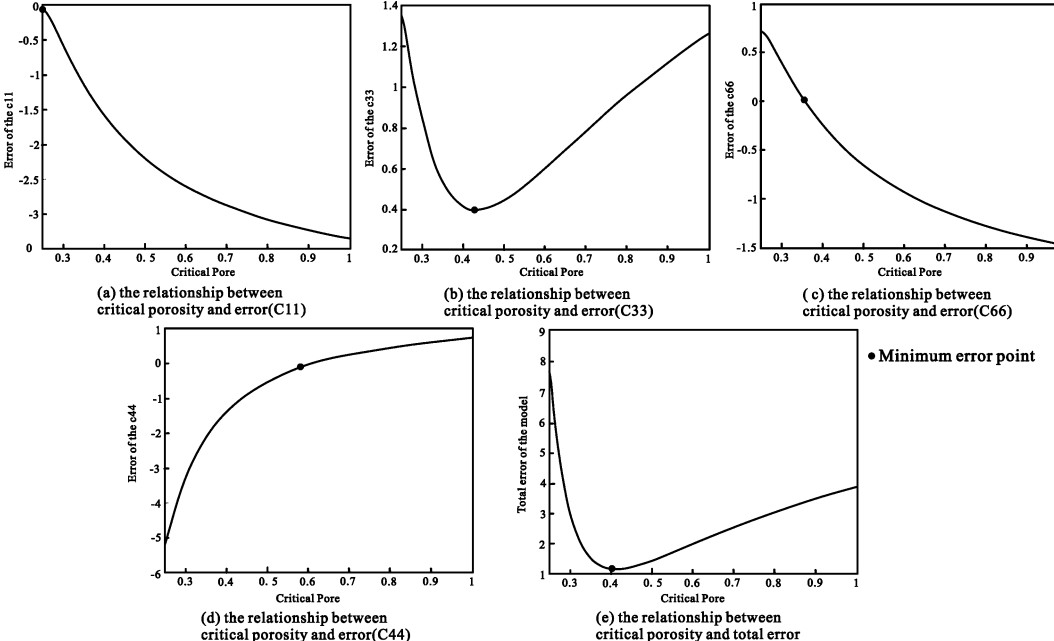

**Figure 15 Stiffness matrix error analysis of the critical porosity. Figure 15a to 15d are the errors between the stiffness coefficients of the core and the stiffness coefficients of the different compaction states (critical porosity). Figure 15e is the error between the stiffness matrix of the core sample and the stiffness matrices of the different compaction states (critical porosity).**

To address this limitation, our approach employs a critical porosity-based method inspired by Bachrach (2011), which effectively approximates shale plate orientation through shale compaction states. This method circumvents the complexities associated with preferred orientation models while calculating the parameters of shale plate orientation. We further validate our models by applying this approach to Bazhenov shale, with detailed shale core data presented in Table 6.

**Table 6 The mineral elastic parameters of the Bazhenov's shale (Vernik and Liu, 1997). The core matrix primarily consists of quartz, feldspar, and clay minerals, while the pores are predominantly filled with organic matter such as kerogen.**

|          | quartz/feldspar | carbonate | clay | Pyrite | Kerogen | porosity | Fluid (Brine) |
|----------|-----------------|-----------|------|--------|---------|----------|---------------|
| Vol (%)  | 46              | 3         | 48   | 3      | 16.8    | 4.12     | N/A           |
| K (GPa)  | 37              | 76.8      | 22.9 | 147.4  | 2.9     | N/A      | 2.2           |
| U (GPa)  | 44              | 32        | 10.6 | 132.5  | 2.7     | N/A      | 0             |

As shown in Figure 15, the critical porosity-based model has the smallest total error at the critical porosity of shale (0.4). This demonstrates that describing the optimal orientation of shale plates based on the shale compaction state is reliable. We further compare the model with other models. The Qian et al.(2014) and our results are shown in Table 7. The optimized shale model has a lower root mean square error than the other widely used shale physical models at the seismic and well logging scales.

**Table 7 the errors of the models. The first row presents the stiffness coefficients calculated from the core measurement data. The last row contains results derived from our rock physics model. The intermediate rows reflect calculations from other scholars' models. The final column indicates the root mean square error between the calculated stiffness matrix and the measured values.**

| Model | C11 (GPa) | C33 (GPa) | C44 (GPa) | C66 (GPa) | Error (RMSE) |
|-------|-----------|-----------|-----------|-----------|--------------|
| Transformed stiffness | 42.38 | 26.23 | 8.68 | 15.23 | 0 |
| Wu et al Result (DEM) | 45.45 | 31.33 | 6.87 | 17.62 | 3.33 |
| Keran Qian Result (SCA+DEM) | 40.93 | 24.48 | 10.07 | 15.75 | 1.36 |
| Keran Qian Result (Backus average) | 42.00 | 22.33 | 9.81 | 16.13 | 2.09 |
| Keran Qian Result (DEM with clay background) | 41.23 | 22.92 | 9.68 | 15.88 | 1.85 |
| Keran Qian Result (DEM with kerogen background) | 42.19 | 23.8 | 10.0 | 16.14 | 1.46 |
| Haoyuan's anisotropic ODF &SCA-DEM model | 43.38 | 26.47 | 7.01 | 16.58 | 1.19 |

## 5.2 Comparison with Hudson models

The widely used model for tight sand is the Hudson model, which effectively describes the elastic characteristics of thin, coin-shaped fractures. Our method further couples the elastic characteristics of thin shale beds with the Hudson model. We used well 5 and 202 from the work area to compare the effects before and after coupling.

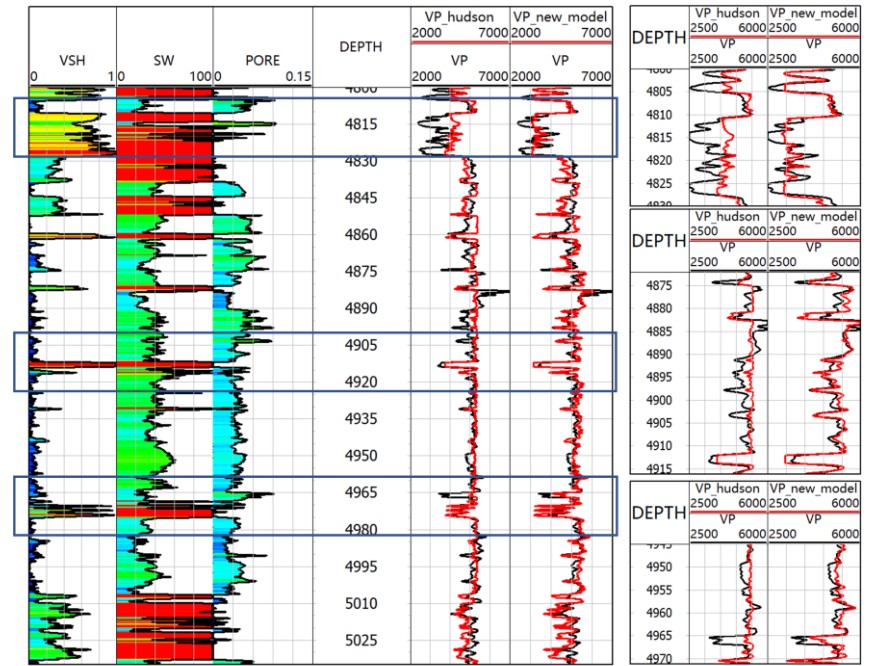

**(a) The comparison of results from two models and well 5**

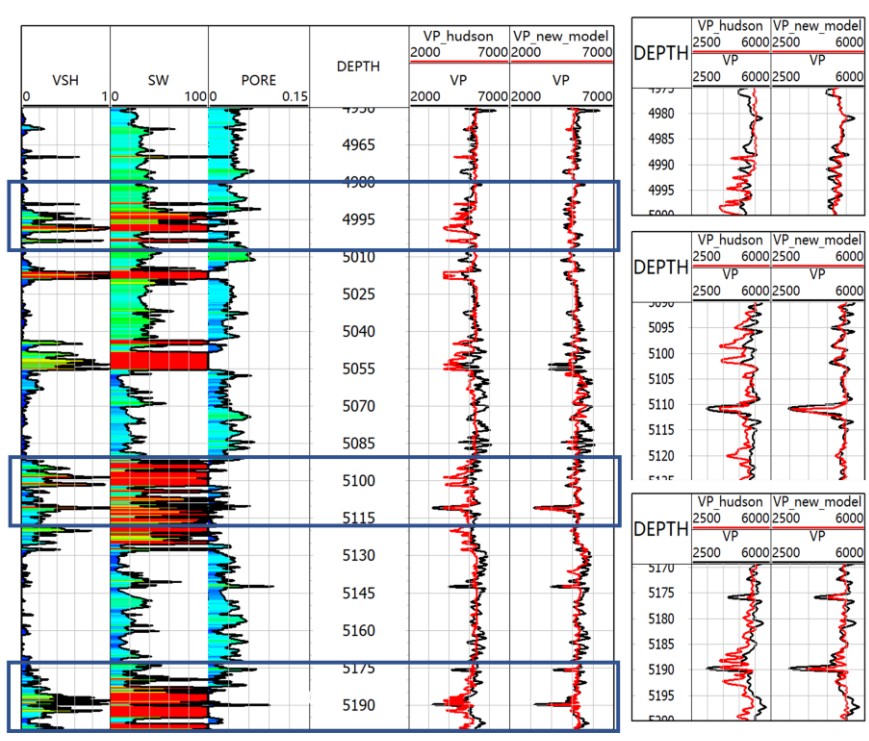

**(b) The comparison of results from two models and well 202**

**Figure 16 the comparison of results from two models and logs. The left side of the figure shows the logging curve (black curve) and the Hudson model (red curve) and the new model results (red curve). The blue box highlights formations where thin shale beds are developed. The subplot on the right displays the fitting details of both models within the corresponding formations.**

The results are shown on the left of Figures 16a and 16b. The three blue boxes refer to the layers that contain both thin shale beds and fractures in the tight sand. Detailed comparison results for these three layers are shown on the right of Figures 16a and 16b. The Hudson model struggles to accurately capture the velocity characteristics of both shale and fractures simultaneously. In contrast, our optimized model fully expresses both the low velocity of the thin shale beds and the fractures in the tight sand. This enables a more accurate representation of layers that simultaneously develop thin shale beds and fractures compared to the Hudson model.

## 5.3 Comparison with other seismic attributes

Before discussing our final seismic attribute results, we first extract some typical seismic attributes that describe tight sand fractures for analysis (Figure 17). The main idea behind these seismic attributes is that fractures enhance seismic wave reflections, resulting in higher amplitude values. Therefore, attributes such as peak amplitude, dominant frequency, reflection energy, composite absolute amplitude, reflection strength energy in dB, and reflection strength slope can indicate the presence of fractures when high. Due to stress concentration near faults, there is considerable consistency between the distribution of faults and fracture development zones. As faults extend from north to south and from west to east within the work area, the fault density and consequently the fracture density increases. It is evident that the seismic response of thin shale beds, which have similar elastic characteristics to fractures, interferes with the identification of fractures using conventional seismic attributes and conventional seismic attributes cannot effectively describe this difference.

When these seismic attributes for the target layer fail to identify fractures, we need to use other methods, such as rock physics model forward and inversion technology. For forward technology, we need to correct our model using a forward model by comparing it with the actual seismic data to obtain reliable fracture parameters, according to the general forward process based on rock physics models. Alternatively, the elastic parameters obtained through seismic inversion techniques may serve as fracture sensitivity parameters in rock physics. Regardless of the method used, the entire technical system requires microstructural parameters such as pore aspect ratio and mudstone plate aspect ratio, which necessitates substantial data support. Additionally, unless there is a simple calculation formula like the Gassmann model, the computation is extensive, with most anisotropic models falling into the latter category. This is a key difficulty in current fracture prediction work.

The new technical process based on the rock physics model proposed in Chapter 4, which utilizes the response characteristics of rock physics analysis to study the target layer through a reference layer, effectively avoids this problem. By analysing the differences between the two layers, we can effectively remove the interference caused by tight sand containing thin shale layers within the target layer.

Moreover, due to stress concentration near faults, there is significant consistency between fault distribution and fracture development zones. The results shown in Figure 14 align more closely with the geological law compared to other conventional seismic attributes.

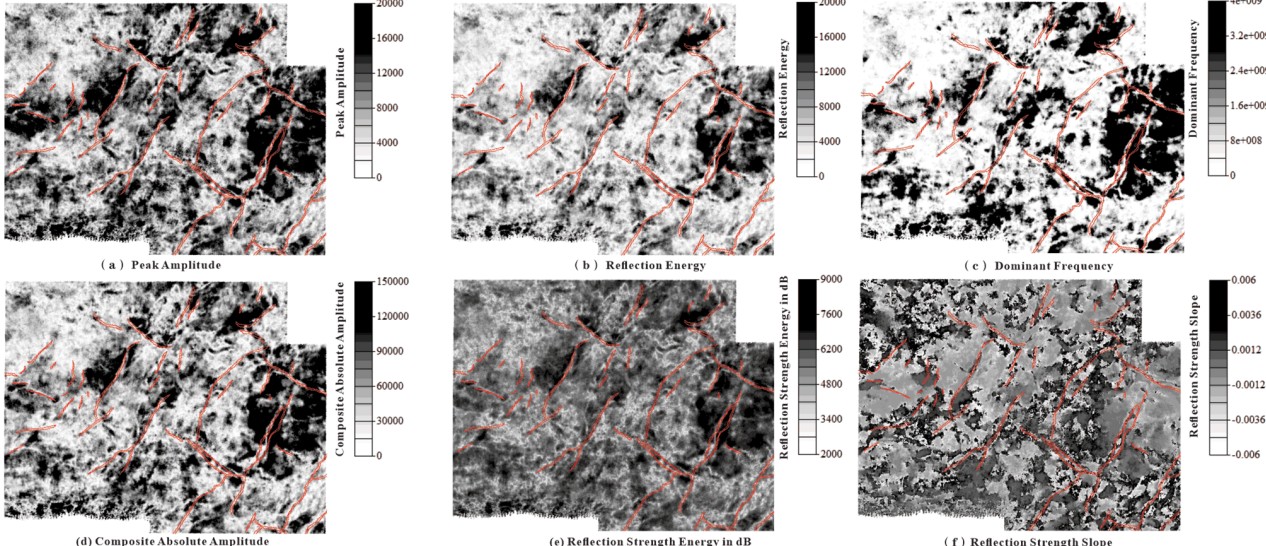

**Figure 17 Commonly used highlight seismic attributes. In the figure, dark areas represent regions with high fracture content, while light and gray areas indicate tight sand or tight sand containing thin shale beds. Red lines mark fault lines.**

### 5.3 The Limitations and Future work

We discuss the method limitation and the potential optimizations from two perspectives. On one hand, there are the optimizations and limitations of the rock physics model for tight sand. On the other hand, there are optimizations and limitations of the seismic attribute extraction techniques based on the rock physics model.

The fracture characterization in the paper is simplified. To fully capture the anisotropy of fractures, a more detailed statistical analysis of fracture dip angles and orientations is required. Fractures with significant differences in dip and orientation should be treated as multiple fracture sets, which can then be superimposed. In these studies, fracture compliance (rather than stiffness) should be the primary focus. We recommend using the LSD model (Michael et al., 1995). Additionally, if fluid flow within fractures is to be studied, fractures and pores across multiple scales must be considered. We suggest adopting the Chapman model(Chapman, 2010). Our regional data show that the area mainly contains horizontal fractures with a common preferred orientation, a sufficiently small aspect ratio, and low fracture porosity and density (Han et al., 2022; Zhang et al., 2022). For our study, which aims to eliminate the VTI anisotropy interference from shale in horizontal fractures, simplifying fractures in the reservoir as a single set of horizontal fractures using the Hudson model is a suitable approach. And the Hudson model, compared to other models, is computationally simpler and requires fewer fracture parameters.

The Brown & Korringa (B&K) formula is derived from the Gassmann model. However, this classical fluid substitution model still has limitations. Thomsen (2022) pointed out that the Gassmann model incorrectly applied an open-system theorem to a closed-system environment, violating the assumptions of undrained compressibility. This results in inconsistencies between

435 Gassmann's results and Biot's original theory. The B&K derivation partially repeats Gassmann's logical error, conflating open and closed systems, particularly when applying unjacketed compression test results to undrained conditions. To optimize fluid substitution models, we suggest two approaches: first, conducting more low-frequency measurement experiments; and second, integrating digital rock techniques to further derive and summarize the elastic changes caused by fluid flow in complex seepage channels.

On the other hand, the technical process of seismic attribute analysis based on the rock physics model is an attempt to follow the theoretical response analysis of the rock physics model. The results show that this approach is better than conventional seismic attributes. However, this is not applicable to all work areas. This requires that the selected reference layer has stable physical properties. If the work area is large, this may not be achievable. Nevertheless, the corresponding research ideas can be further expanded. We can use the seismic response of tight sand forward modelling, incorporating specific factors such as

thin shale layers and cracks, as a waveforms dictionary. Then, we can use increasingly mature artificial intelligence technology to match the actual seismic response to more accurately and quantitatively explain the details behind the waveform.

## 6 Conclusion

This study focuses on the previously neglected thin shale beds within tight sand below the log observation scale. Our rock physics model analysis demonstrates that both fractures and thin shale beds within tight sand influence dynamic elastic

parameters similarly. We found that porosity significantly affects the elastic and anisotropy parameters of fractures more than it does for thin shale beds. However, subsequent forward modelling further demonstrated that certain tight sands containing fractures exhibit similar responses to those containing thin shale beds. Therefore, relying solely on fracture models can result in errors for tight sands containing thin shale beds. To address this issue, our new model integrates two types of anisotropy: fracture anisotropy and shale anisotropy models. The new model achieves better results in tight sands with thin shale beds

compared to the Hudson model.

In our rock physical analysis, we found that pre-stack seismic data could distinguish between fractures and thin shale beds. However, in practical workflows, post-stack seismic data is more widely used due to its convenience. To identify fractures using post-stack data, we further analyzed the physical properties of this complex sand, significantly simplifying the workflow. Fortunately, complex sand has insufficient porosity. We also found that the greater the porosity, the more distinct the difference

between fractures and thin shale beds. As a result, we considered the aspect ratio of the waveform as a new seismic attribute and applied it to the region. The results show that the new attribute aligns closely with fault distribution and can effectively characterize fracture distribution in tight sand.

Our application and analysis demonstrate the effectiveness of the developed hybrid rock physics model in identifying thin shale beds and fractures, with advantages over conventional methods. However, our model has limitations, including the dip of fractures, the presence of organic matter within shale, and other microscopic factors that may affect its accuracy in describing the elastic parameters of specific fractures and shales. Future research should conduct microscopic experiments on this type of tight sand, optimize the model, enhance its adaptability to tight sands with different microstructures, and explore more practical application scenarios to enhance the potential and practical significance of the research results.

**Author contributions.**

HL was responsible for the writing, analysis, discussion, and conclusion of the paper. XH contributed to the review and revision of the paper. LF, LL, and TC were responsible for data collection, organization, and conceptual support.

**Competing interests**

The contact author has declared that none of the authors has any competing interests.

**Acknowledgements.**

We appreciate the constructive comments from the two anonymous referees and the editors, Florian Fusseis and Charlotte Krawczyk, which have significantly improved the manuscript. In addition, we would like to thank Professor Limin Sa for providing the authors with valuable suggestions regarding fracture research.

**Financial support.**

This research was supported by the National Natural Science Foundation of China (grant no. U20B2016), the Natural Science Foundation of Sichuan Province (grant no. 2024NSFSC1992), and the Sinopec Science and Technology Research and Development Project (grant no. P2158).

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
