# Peer review of "Elastic anisotropy differentiation of thin shale beds and fractures using a novel hybrid rock physics model"

_EGUsphere, 2024_

## Author Response (AR1)

We would like to express our sincere gratitude to the two reviewers (one major revision and one minor revision.) and the editor for their valuable feedback during the peer review process. We provide a summary of the reviews' opinions and our detailed responses to each comment below, along with a description of the revisions made to the manuscript. The manuscript specifically addresses the following concerns raised by the reviewers:

- Some citations and equation formatting are not accurate.
- The description of some rock physics models requires revision.
- Inadequate emphasis on the assumption of horizontal fractures, which may lead to potential controversies.
- Insufficient geological background information, particularly concerning horizontal fractures.

In response to the reviewers' comments, we have made the following revisions and improvements in the new version of the manuscript:

- We carefully corrected the equations according to the references and unified the variables in the manuscript for better normalization.
- In Chapters 2 and 3, we have supplemented and revised the rock physics models to address the lack of clarity in the modeling process, providing additional explanations on their applicability. Furthermore, we included additional discussions in the discussion section.
- We analyzed the horizontal fractures in the geological introduction part and the whole manuscript is based on the VTI assumption. We supplemented the assumption in the introduction, modeling, background, and discussion sections.
- We expanded Section 3.1 with a detailed geological background of horizontal fractures in the study area, including relevant references.

The specific responses and revisions for each comment are as follows (Brown is the Comment, Blue is the response, and Black is the specific modification content.):

Reviewer 1:
We sincerely appreciate the feedback from Reviewer 1. After multiple rounds of discussions, we categorized all comments from Reviewer 1 based on the suggestions in the final comment to facilitate our responses. (Brown is the Comment, Blue is the response, and Black is the specific modification content.):

Comment 1:
The paper must emphasize that the subject fractures are bed-parallel fractures with assumed VTI symmetry. The more common tectonic fractures may be differentiated from thin beds via their orientation alone, with no modelling required.

- The primary means to differentiate thins beds from fractures is through their different orientations, which is imbedded here but not emphasized. No elaborate rock physics model, containing many strong assumptions, is required.
- More fundamentally, however: All that is needed to differentiate cracks from thin-bedded shales is to recognize that the shales have VTI symmetry, and the cracks do not. No elaborate modelling is required.
- The authors have also employed an unnecessarily restrictive model for fractures, one that assumes "penny-shaped" cracks. That makes for HTI, if embedded in an otherwise isotropic matrix (not if imbedded in shales, or in thin-beds). If, for example, the fractures are "ribbon-shaped", i.e. if they are joints, the HTI model is not accurate. HTI is a special case of azimuthal anisotropy, probably never occurring in nature. A more general model for cracks embedded in shales or thin-beds is given by Sayers, C. M., 2022, Elastic properties of fractures in transversely isotropic media: Journal of Applied Geophysics, 197.
- More fundamentally, however: All that is needed to differentiate cracks from thin-bedded shales is to recognize that the shales have VTI symmetry, and the cracks do not. No elaborate modelling is required.

Response 1:
Thanks for your comments, it is a very interesting and important question. Normally, the vertical fractures and horizontal thin shales, generate anisotropy with different orientations, do not need extra rock physics model. However, in our working area, horizontal (rather than vertical) fractures exit and generate the same type of anisotropy compared with shale beds. And this makes the characterization of fractures becoming difficult. Our work actually wants to differentiate the anisotropy caused by horizontal fractures and shale beds. We understand the misunderstanding lies in our original manuscript and we have revised the manuscript and added the Section 3.1 to make it clearer.

Revision 1:
- Replace some "fracture" in the manuscript to "Horizontal fracture."
- Added the following to the Introduction Chapter:" Based on the fracture orientations and dip angles within the study area, we assume the fracture anisotropy to be VTI anisotropy."
- In the methods section, we replaced "To construct a rock physics hybrid model that can effectively represent both the anisotropy of thin shale beds and horizontal fractures, we

proposed the modelling procedure shown in Figure 1." with "To study the VTI anisotropy of shale and horizontal fractures, we proposed the modeling workflow shown in Figure 1. (Vertical fractures, due to their orientation, exhibit HTI anisotropy, which can be directly distinguished from shale without modeling.)"

- We have added the following emphasis in the background section of Chapter 3:

"It revealed that these horizontal fractures will exhibit VTI anisotropy. Therefore, we use the VTI equivalent model to study the anisotropy of fractures in the area."

Comment 2:
The field occurrence of subsurface bed-parallel fractures must be amply demonstrated with cited references and discussion. Outcrops are not acceptable proof, since the near-surface has low vertical stress.

- Please provide a reference which shows the field occurrence of horizontal fractures with VTI symmetry.

Response 2:
We have uploaded images of cores and imaging logs that demonstrate the horizontal fractures in this area. These images are from other scholars' studies. We have added Section 3.1 to provide a detailed introduction to horizontal fractures and have highlighted horizontal fractures as much as possible throughout the text. This section includes our existing core and imaging logging data on horizontal fractures. Additionally, we referenced a large number of studies to support our explanation.

Revision2:
We have added a detailed description of the region in Section 3.1:

**"3.1 Geological background**

[revised manuscript text omitted]

"

Comment 3: The bed-parallel fractures may or may not have VTI symmetry since their orientation alone is not sufficient to establish that (as it is with the bedding). Logs establish orientation, but not symmetry. The shear slip on these fractures may or may not destroy the VTI symmetry. So, VTI symmetry of the fractures must be explicitly assumed.

Response 3: This is a very meaningful question. Our assumption is based on the fracture orientations from imaging logging statistics and the fracture dip angles of the cores presented in Section 3.1. We further emphasized this assumption throughout the manuscript. This assumption is also used by other researchers in the same region (Han et al., 2022; Zhang et al., 2022).

Revision3:
- Added the following to the Introduction Chapter:"Based on the fracture orientations and dip angles within the study area, we assume the fracture anisotropy to be VTI anisotropy."
- We have added the following emphasis in the background section of Chapter 3:
  "It revealed that these horizontal fractures will exhibit VTI anisotropy. Therefore, we use the VTI equivalent model to study the anisotropy of fractures in the area."
- We have discussed this assumption in the discussion section:
  "Our regional data show that the area mainly contains horizontal fractures with a common preferred orientation, a sufficiently small aspect ratio, and low fracture porosity and density (Han et al., 2022; Zhang et al., 2022). For our study, which aims to eliminate the VTI anisotropy interference from shale in horizontal fractures, simplifying fractures in the reservoir as a single set of horizontal fractures using the Hudson model is a reasonable approach.

Comment 4: The modeling and discussion must respect the criticisms stated earlier.

- Even deeper, the authors have not mastered the prior literature, for example they have misunderstood and mischaracterized Hill (1952), Brown and Korringa (1975), Hudson (1980), Thomsen (2023) among others.
- Hill (1952): The Voigt and Reuss "limits" discussed by Hill are only valid for mixtures of isotropic minerals, whereas all of the minerals included in this ms are anisotropic.
- Brown and Korringa (1975): In the present ms, equation (3) has two errors:

    In the second term on the right, the compliance elements with superscript "sand" should instead have superscript "dsand"; this may be just a typo.

    In the same term, the compliance elements s with superscript "sand0" should instead have superscript "sandM", and the stiffness element K with subscript "sand0" should instead have subscript "sandM". This notation more closely follows that of Brown and Korringa. Many authors since 1975 (including Mavko et al) have considered M to represent the solid Mineral of the porous rock, but as explained by Thomsen (2023), a better interpretation of M is Mean.

- Thomsen (2023): This paper revises the treatment of fluids within porous rocks in a fundamental way, not consistent with Mavko et al (2020)
- Physically, their first error occurs at equation (2) (which does not appear in Hudson, 1980). In an aggregate like that considered, the compliances of the members are additive, not their stiffnesses (see e.g., Schoenberg and Sayers, Geophysics, 60, 204-211, 1995).
- Hudson (1980,1981): These papers consider only cracks in non-porous rock, like granite. If the rock also has non-crack porosity, hydraulically connected to the cracks (e., if they are typical sedimentary rocks), the authors should instead cite Hudson et al, Geophys. J. Int. (1996) 124, 105-112.
- The references cited in AC1, by Hudson (1981) and by Mavko et al (2020) both state explicitly that these formulae are for small perturbations only. They are first order Taylor approximations of the correct formula, which is given by Schoenberg and Sayers (1995).

Response 3: We appreciate the reviewer's careful reading. Some of the questions are partially repetitive and we concluded to three main issues: the Hill problem, the Hudson problem, and the Brown and Korringa model problem (Thomsen also pointed out this issue in 2023).

Hill model: We use the bulk modulus and shear modulus of minerals listed in Table 2, which can be combined using the Hill formula. The formula for converting the moduli calculated using the Hill formula into the stiffness matrix should not be omitted. We have added Equations (1) and (2). Through this revision, we have provided a detailed explanation of how the Hill formula is used to calculate the bulk modulus and shear modulus of the synthesized mineral matrix, and how these moduli are then converted into an isotropic background stiffness matrix.

Hudson model: We sincerely appreciate the reviewers' comments, which have helped us gain a deeper understanding of the Hudson model. we have updated the references to Hudson (1996). The fracture porosity in the study area is low, making it well-suited for Hudson's small perturbation theory. In Chapter 3, we have supplemented an explanation of Hudson's theory (Hudson's model

treats fractures as small perturbations to the background stiffness, assuming that fractures induce only minor changes to the medium. Specifically, it employs a first-order Taylor expansion to approximate the impact of fractures on the stiffness tensor.) and elaborated on the rationale for its selection.

Brown and Korringa model: (6)    Thanks for the suggestions, we standardized the expression for the solid mineral and changed all footnotes from "0" to "m.". Although Thomsen pointed out that there is a logic error in Gassmann's assumption, Gassmann equation is widely used by researchers and proved to be valid. We also shown that Gassmann equation is effective in our working area by compare the theoretical curve with logging data (figure 6), thus we believe that Gassmann equation works in our area and the logic error will not have impact on our results.

Revision 3:
- We revised the description of the Hill model and supplemented it with the formula for converting bulk and share modulus into the stiffness matrix form:
  "For the fracture skeleton, we used the VRH model (Hill, 1952) to build the sand matrix (*Msand*):

$$Msand = \left( \sum_{i=1}^{N} f_i M_i + \frac{1}{\sum_{i=1}^{N} \frac{f_i}{M_i}} \right) / 2, \tag{1}$$

  where $f_i$ is the content of the i-th mineral in the matrix, $M_i$ is the isotropic modulus of the rock. *Msand* is background isotropic modulus (bulk modulus $Km$ or shear modulus $\mu m$). The isotropic modulus can be converted into the sand isotropic stiffness of the matrix using the following equation:

$$\boldsymbol{C}_{sandm} = \begin{bmatrix} Km + \frac{4}{3}\mu m & Km - \frac{2}{3}\mu m & Km - \frac{2}{3}\mu m & 0 & 0 & 0 \\ Km - \frac{2}{3}\mu m & Km + \frac{4}{3}\mu m & Km - \frac{2}{3}\mu m & 0 & 0 & 0 \\ Km - \frac{2}{3}\mu m & Km - \frac{2}{3}\mu m & Km + \frac{4}{3}\mu m & 0 & 0 & 0 \\ 0 & 0 & 0 & \mu m & 0 & 0 \\ 0 & 0 & 0 & 0 & \mu m & 0 \\ 0 & 0 & 0 & 0 & 0 & \mu m \end{bmatrix}, \tag{2}"$$

- We updated the Hudson model:
  "Hudson model (Hudson, 1980) is based on a scattering-theory analysis of the mean wave field in an elastic solid with thin, penny-shaped ellipsoidal cracks or inclusions. He proposed a Taylor expansion approximation to calculate the stiffness matrix for fracture-porosity composite system (Hudson et al., 1996):

$$C_{dsand} = C_{sandm} + C^1 + C^2 , \tag{3}$$

  where $C^1$ is the first-order correction term for the anisotropy caused by the fracture, and $C^2$ is the second-order correction term of the anisotropy caused by the mutual coupling between the directional fractures. Both $C^1$ and $C^2$ are calculated from the pore aspect ratio a, the rock matrix porosity $\phi m$ and fracture porosity $\phi f$ ($\phi = \phi m + \phi f$). We assume that the percentage of fracture porosity to total porosity $\phi$ in the reservoir is a constant (3.2%). These parameters of skeleton were obtained from log curves and thin sections. "
- We supplemented the calibration process of the Hudson model in Chapter 3 to demonstrate its

applicability to the horizontal fractures in the study area:

"The compliance effectively characterizes a material's ability to deform under stress, making it particularly suitable for representing fractures. The Linear Slip Deformation (LSD) model proposed by Schoenberg and Sayers(1995) assumes a linear relationship between the displacement discontinuity across a fracture surface and the applied stress, enabling the compliance contributions from multiple fractures to be directly summed. In contrast, Hudson's model treats fractures as small perturbations to the stiffness tensor, based on the assumption that fractures introduce only minor modifications to the medium. Specifically, it uses a first-order Taylor expansion to approximate the effects of fractures on the stiffness. This approach makes stiffness perturbation a more suitable and computationally efficient framework for modeling low fracture densities. Furthermore, stiffness-based models offer a more direct approach for analyzing seismic wave propagation, particularly when fracture porosity is sufficiently low. When fracture porosity exceeds 0.45% (fracture density = 0.1), the Hudson model becomes less effective in describing the elastic properties of fractures (Figure 5). In this study, the fractures exhibit a maximum porosity of 0.39%, which is well within the descriptive capabilities of the Hudson model. Therefore, we adopted the Hudson model for our analysis.

[Figure]

Figure 5 The solid line represents the first-order Hudson formula, the dashed line represents the second-order Hudson formula, and the points indicate well log samples. The first-order results of the Hudson model consider the effects of fractures, while the second-order results account for both the effects of fractures and their interactions. In this study, we used the second-order results of the Hudson model."

- We have revised the BK model and other sections, updating the subscript for the matrix from "0" to "m".We have revised the BK model and other sections to update the subscript for the matrix from "0" to "m." The corrected expression for the BK model is as follows:

$$\text{``}s_{ijkl}^{sand} = s_{ijkl}^{dsand} - \frac{\left(s_{ijaa}^{\text{sand}}-s_{ijaa}^{sandm}\right)\left(s_{bbkl}^{\text{sand}}-s_{bbkl}^{sandm}\right)}{\left(s_{ccdd}^{\text{sand}}-s_{ccdd}^{sandm}\right)-\phi\left(\frac{1}{K_{\text{fl}}}-\frac{1}{K_{sandm}}\right)}\text{,}\tag{4}$$

where the parameters $s_{ijkl}^{dsand}$ and $s_{ijkl}^{sandm}$ represent the flexibility of dry rock skeleton and rock matrix minerals respectively. The stiffness matrix can be inverted from flexibility matrix following $C_{ijkl}\,S_{ijkl} = I$ and vice versa. $K_{\text{fl}}$ can be obtained by the Wood (1956) formula."

- We have added Thomsen's discussion on the BK issue in the discussion section:

"Although the Brown & Korringa (B&K) formula is derived from the classical fluid substitution model, the Gassmann model, it is worth noting that this fluid model still has limitations. Thomsen (2022) pointed out that the Gassmann model incorrectly applied an open-system theorem to a closed-system environment, violating the assumptions of undrained

compressibility. This results in inconsistencies between Gassmann's results and Biot's original theory. The B&K derivation partially repeats Gassmann's logical error, conflating open and closed systems, particularly when applying unjacketed compression test results to undrained conditions.To optimize fluid substitution models, we suggest two approaches: first, conducting more low-frequency measurement experiments; and second, integrating digital rock techniques to further derive and summarize the elastic changes caused by fluid flow in complex seepage channels. These methods aim to address the limitations of fluid effects in the modelling process presented in this study."

Comment 1: Please check the table numbering in the manuscript.

Response 1: We corrected the issue with missing Table 2 and thoroughly checked the sequence of table numbers.

Revision 1: We have revised the table numbers to ensure their correct sequence.

Comment 2: In Figure 1, the Hudson model is used to compute characteristics of fractured sandstone. Please discuss how this method accounts for matrix porosity or clarify the assumptions in the modeling.

Response 2: We apologize for not adequately emphasizing the relationship and assumptions between matrix porosity and fracture porosity in the text. We considered matrix porosity as part of the matrix, while the elastic effects of fracture porosity and its interaction with matrix porosity on sand were calculated using the original Equation 2 (with the complete calculation formulas detailed in Hudson's 1996 paper). We have updated the explanation of the Hudson model. Additionally, the ratio of fracture porosity to total porosity was set as a constant (3.2%) based on the statistics of the study area.

Revision 2:

- We have added an explanation of matrix porosity and fracture porosity in the context of Hudson modeling.:

  "Hudson model (Hudson, 1980) is based on a scattering-theory analysis of the mean wave field in an elastic solid with thin, penny-shaped ellipsoidal cracks or inclusions. He proposed a Taylor expansion approximation to calculate the stiffness matrix for fracture-porosity composite system (Hudson et al., 1996):

  $$C_{dsand} = C_{sandm} + C^1 + C^2 \,, \tag{5}$$

  where $C^1$ is the first-order correction term for the anisotropy caused by the fracture, and $C^2$ is the second-order correction term of the anisotropy caused by the mutual coupling between the directional fractures. Both $C^1$ and $C^2$ are calculated from the pore aspect ratio a, the rock matrix porosity $\phi m$ and fracture porosity $\phi f$ ($\phi = \phi m + \phi f$). We assume that the percentage of fracture porosity to total porosity $\phi$ in the reservoir is a constant (3.2%). These parameters of skeleton were obtained from log curves and thin sections."

- In Chapter 3, we introduced the analysis of the proportion of fracture porosity and matrix porosity in the reservoir:

  "We conducted statistical analysis on the average fracture porosity and total porosity interpretation results from 10 wells in the study region. The results are shown in the figure 4. The fracture porosity in the region ranges from 0.04% to 0.39%, accounting for 3.2% of the total porosity.

[Figure]

**Figure 4 Fracture porosity and matrix porosity analysis. The bar chart represents the average values of porosity and fracture porosity within the target interval from logs in the area. The line chart indicates the percentage of fracture porosity relative to total porosity.**

"

Comment 3: Please explain the meaning of the "shale domain" in Figure 1.

Response 3: The "shale domain" is based on Bandyopadhyay (2008). We will include the reference and explanation, specifically referring to the equivalent unit slabs of shale plates.

Revision 3: We have revised the original text, changing "shale domain" to "clay domain," and have added references for the clay domain:

"The thin shale beds in the tight sand can be considered as composed of shale domain clay domains (Bandyopadhyay, 2008)."

Comment 4: In Figure 4, please clarify the meaning of porosity. Does it specifically refer to fracture porosity?

Response 4: The porosity shown in the new manuscript figure 8 represents total porosity. In Chapter 4, we analyze the VTI anisotropy of shale and horizontal fractures. Fracture porosity alone cannot account for the VTI anisotropy of shale. we discussed the impact of fracture porosity on the framework in Chapter 3.

Revision 4: We have added the analysis of fracture porosity in Chapter 3:

[Figure]

"

**Figure 5 The solid line represents the first-order Hudson formula, the dashed line represents the second-order Hudson formula, and the points indicate well log samples. The first-order results of the Hudson model consider the effects of fractures, while the second-order results account for both the effects of fractures and**

**their interactions. In this study, we used the second-order results of the Hudson model.”**

Comment 5: In Figure 8, consider using the term "anisotropy aspect ratio." For example, why call it "anisotropy"? Also, "aspect ratio" is typically used to describe pore or fracture geometry.

Response 5: We revised the attribute name to "fracture and shale ratio" to make it easier to understand.

Revision 5: We revised the attribute name to "fracture and shale ratio" to make it easier to understand.

Comment 6: Please provide more geological descriptions of bed-parallel fractures in the study area.

Response 6: We expanded both the introduction and Section 3.1 with more detailed geological descriptions of fractures.

Revision 6: We have supplemented the explanation of horizontal fractures in Chapter 3, including references to core samples, imaging logging, and other related studies:

**“3.1 Geological background**

[revised manuscript text omitted]

---

## Author Response (AR2)

Dear Editors

We sincerely appreciate your valuable suggestions and have revised the manuscript accordingly. Upon careful examination, we identified that the absence of tracked changes was due to the use of EndNote software for inserting references. To address this, we have provided a detailed list specifying the locations and citations of the newly added references.

Furthermore, after consulting with the editorial office and receiving their approval:

- We have followed Professor Limin Sa's wishes by removing his name from the author list and acknowledging his contributions in the acknowledgment section. In line with this change, the author list, contributions and acknowledgment sections have been updated accordingly in the revised version.
- We have updated the acknowledgments and financial support sections in the revised version.

**Introduction of New References:**

**The locations where the newly added references have been cited are as follows:**

These compacted shales exhibit a microstructure with a preferential orientation in the plane (Bandyopadhyay, 2008)

Chapman et al. (2010) gave a multi-scale fractures equivalent model.

Our regional data show that the area mainly contains horizontal fractures with a common preferred orientation, a sufficiently small aspect ratio, and low fracture porosity and density (Han et al., 2022; Zhang et al., 2022).

The research focuses on the tight sand gas in the Xujiahe Formation in the Sichuan Basin. The region has dense lithology, with fractures serving as the primary migration pathways (Huang et al., 2022).

He proposed a Taylor expansion approximation to calculate the stiffness matrix for fracture-porosity composite system (Hudson et al., 1996):

In these studies, fracture compliance (rather than stiffness) should be the primary focus. We recommend using the LSD model (Michael et al., 1995).

Previous Chinese scholars have analysed horizontal fractures, which exhibit preferential distribution and structural features of VTI anisotropy (Su, 2011).

Horizontal fractures can help identify tight gas reservoirs in this area (Yue et al., 2018; Zhang, 2021).

The developed horizontal fractures can also effectively assist in the water injection development of tight gas (Zhao et al., 2021).

**List of Newly Added References:**

Bandyopadhyay, K.: Seismic anisotropy Geological causes and its implications to reservoir geophysics, Stanford University, 2008.

Chapman, M.: Frequency-dependent anisotropy due to meso-scale fractures in the presence of equant porosity, Geophysical Prospecting, 51, 2010.

Han, L., Liu, J., Yang, R., Zhang, G., and Zhou, Y.: Application of Prestack Elastic Impedance Inversion Method Based on VTI Media: A Case Study of Tight Sandstone Fractured Reservoirs in the Xujiahe Formation, Oil & Gas Reservoir Evaluation and Development, 12, 313-319, 328, 10.13809/j.cnki.cn32-1825/te.2022.02.006, 2022.

Huang, Y., Wang, A., Xiao, K., Lin, T., and Jin, W.: Types and genesis of sweet spots in the tight sandstone gas reservoirs: Insights from the Xujiahe Formation, northern Sichuan Basin, China, Energy Geoscience, 3, 270-281, 2022.

Hudson, J. A., Liu, E., and Crampin, S.: The mechanical properties of materials with interconnected cracks and pores, Geophysical Journal International, 1, 1996.

Michael, Schoenberg, Colin, M., and Sayers: Seismic anisotropy of fractured rock, GEOPHYSICS, 60, 204-211, 1995.

Su, Y.: Fracture Identification and Distribution Evaluation in the Xu-2 Member of the Xinchang Area, Chengdu University of Technology, 2011.

Yue, D., Wu, S., Xu, Z., Xiong, L., Chen, D., Ji, Y., and Zhou, Y.: Reservoir quality, natural fractures, and gas productivity of upper Triassic Xujiahe tight gas sandstones in western Sichuan Basin, China, Marine & Petroleum Geology, 89, 370-386, 10.1016/j.marpetgeo.2017.10.007, 2018.

Zhang, J., Fan, Xin, Huang, Zhiwen, Liu, Zhongqun, Qi, Yuanchang: Evaluation Method of Anisotropic In-Situ Stress in the Upper Triassic Xujiahe Formation Reservoir in the Western Sichuan Depression, Sichuan Basin, Oil & Gas Geology, 42, 963-972, 10.11743/ogg20210416, 2021.

Zhao, H., Shang, X., Li, M., Zhang, W., Wu, S., Lian, P., and Duan, T.: Investigation on petrophysical properties of fractured tight gas sandstones: a case study of Jurassic Xujiahe Formation in Sichuan Basin, Southwest China, Arabian Journal of Geosciences, 14, 1-8, 2021.

**Update Information:**

**Author contributions.**

HL was responsible for the writing, analysis, discussion, and conclusion of the paper. XH contributed to the review and revision of the paper. LF, LL, and TC were responsible for data collection, organization, and conceptual support.

**Acknowledgements.**

We appreciate the constructive comments from the two anonymous referees and the editors, Florian Fusseis and Charlotte Krawczyk, which have significantly improved the manuscript. In addition, we would like to thank Professor Limin Sa for providing the authors with valuable suggestions regarding fracture research.

**Financial support.**

This research was supported by the National Natural Science Foundation of China (grant no. U20B2016), the Natural Science Foundation of Sichuan Province (grant no. 2024NSFSC1992), and the Sinopec Science and Technology Research and Development Project (grant no. P2158).